# Shared Common Ancestry of Rodent Alphacoronaviruses Sampled Globally

**DOI:** 10.3390/v11020125

**Published:** 2019-01-30

**Authors:** Theocharis Tsoleridis, Joseph G. Chappell, Okechukwu Onianwa, Denise A. Marston, Anthony R. Fooks, Elodie Monchatre-Leroy, Gérald Umhang, Marcel A. Müller, Jan F. Drexler, Christian Drosten, Rachael E. Tarlinton, Charles P. McClure, Edward C. Holmes, Jonathan K. Ball

**Affiliations:** 1School of Life Sciences, University of Nottingham, Nottingham NG7 2UH, UK; t.tsoleridis@nottingham.ac.uk (T.T.); nixjc1@exmail.nottingham.ac.uk (J.G.C.); okechukwu.onianwa@phe.gov.uk (O.O.); patrick.mcclure@nottingham.ac.uk (C.P.M.); 2School of Veterinary Medicine and Science, University of Nottingham, Sutton Bonington Campus, Loughborough LE12 5RD, UK; rachael.tarlinton@nottingham.ac.uk; 3Wildlife Zoonoses and Vector-borne Diseases Research Group, Animal and Plant Health Agency (APHA), Weybridge-London KT15 3NB, UK; Denise.Marston@apha.gsi.gov.uk (D.A.M.); Tony.Fooks@apha.gsi.gov.uk (A.R.F.); 4Anses, Laboratoire de la rage et de la faune sauvage, 54220 Malzéville, France; elodie.monchatre-leroy@anses.fr (E.M.-L.); gerald.umhang@anses.fr (G.U.); 5Charité-Universitätsmedizin Berlin, Corporate Member of Freie Universität Berlin, Humboldt-Universität zu Berlin, and Berlin Institute of Health, Institute of Virology, 10117 Berlin, Germany; marcel.mueller@charite.de (M.A.M.); felix.drexler@charite.de (J.F.D.); christian.drosten@charite.de (C.D.); 6Marie Bashir Institute for Infectious Diseases and Biosecurity, Charles Perkins Centre, School of Life and Environmental Sciences and Sydney Medical School, The University of Sydney, Sydney, NSW 2006, Australia; edward.holmes@sydney.edu.au

**Keywords:** coronavirus, alphacoronavirus, rodents, ancestry, recombination, evolution

## Abstract

The recent discovery of novel alphacoronaviruses (alpha-CoVs) in European and Asian rodents revealed that rodent coronaviruses (CoVs) sampled worldwide formed a discrete phylogenetic group within this genus. To determine the evolutionary history of rodent CoVs in more detail, particularly the relative frequencies of virus-host co-divergence and cross-species transmission, we recovered longer fragments of CoV genomes from previously discovered European rodent alpha-CoVs using a combination of PCR and high-throughput sequencing. Accordingly, the full genome sequence was retrieved from the UK rat coronavirus, along with partial genome sequences from the UK field vole and Poland-resident bank vole CoVs, and a short conserved ORF1b fragment from the French rabbit CoV. Genome and phylogenetic analysis showed that despite their diverse geographic origins, all rodent alpha-CoVs formed a single monophyletic group and shared similar features, such as the same gene constellations, a recombinant beta-CoV spike gene, and similar core transcriptional regulatory sequences (TRS). These data suggest that all rodent alpha CoVs sampled so far originate from a single common ancestor, and that there has likely been a long-term association between alpha CoVs and rodents. Despite this likely antiquity, the phylogenetic pattern of the alpha-CoVs was also suggestive of relatively frequent host-jumping among the different rodent species.

## 1. Introduction

Among the viruses that infect both humans and animals, coronaviruses are common and important pathogens. The first coronavirus (CoV) was discovered during the 1930s [1], and the first human coronavirus was characterized in the 1960s [2]. Since then, five novel human coronaviruses have been discovered. The coronavirus family gained particular notoriety with the emergence of the severe acute respiratory syndrome (SARS) in 2002/2003 and the Middle East respiratory syndrome (MERS) in 2013. In humans, coronaviruses are associated with respiratory disease, causing a range of symptoms from mild common colds to more severe lower respiratory tract infections that can be lethal. CoV infection in a range of non-human mammals and birds is associated with enteric and respiratory diseases as well as hepatitis and neurological disorders [3].

Rodents are an important source of emerging virus infections [4]. Rodentia is the single largest mammalian order and comprises ~40% of all mammals [5], including ~2200 species such as rats, mice, and voles [6]. Despite the key role played by rodents in virus emergence, until recently, only two rodent CoVs had been characterized—murine hepatitis virus (MHV) [7] and rat sialodacryoadenitis coronavirus (SADV) [8], both of which are from the same viral species and are betacoronaviruses.

We previously reported the discovery of novel alpha-CoVs in European rodents and shrews, encompassing several species such as field voles (*Microtus agrestis*), bank voles (*Myodes glareolus*), and rats (*Rattus norvegicus*) [9]. Since then, additional alpha-CoVs have been described infecting a variety of Chinese rodents, including field mice (*Apodemus chevrieri*), grey red-backed voles (*Myodes rufocanus*), and lesser rice field rats (*Rattus losea*) [10,11]. Interestingly, a recently discovered rabbit alphacoronavirus also clustered with rodent alpha-CoVs [12].

To gain a better understanding of the evolution of alphacoronaviruses in rodents, particularly how long these virus-host associations have been established, we recovered and characterized complete or partial genomes for European Norway rat, field vole, and bank vole CoVs, and compared these sequences to those available for other rodent CoVs obtained in other parts of the world. Our new phylogenetic analyses provide compelling evidence that the alpha-CoVs sampled from rodents to date have arisen from a single common ancestor, suggestive of a long-term virus-host association.

## 2. Materials and Methods

### 2.1. Samples

Samples previously reported to be positive for alpha-CoVs were used in this study [9,12]. These samples included a UK *Rattus norvegicus* (UK*Rn*3), UK *Microtus agrestis* (UK*Ma*1), Poland-resident *Myodes glareolus* (PL*Mg*1), and the France-resident *Oryctolagus cuniculus* (L232) (Table 1).

### 2.2. Nucleic Acid Preparation

Total RNA was extracted from approximately one cubic millimetre sections of liver (UK*Rn*3, PL*Mg*1) or intestinal tissue samples (UK*Ma*1) using the GenElute™ Mammalian Total RNA Miniprep Kit (Sigma Aldrich, Steinheim, Germany). Total RNA was extracted from intestinal fluid for the France resident L232 rabbit CoV using the QIAamp Viral RNA Mini Kit (Qiagen, Hilden, Germany). The RNA to cDNA EcoDry™ Premix - Random Hexamers kit (Clontech, Saint-Germain-en-Laye, France) was used for cDNA synthesis.

### 2.3. High-Throughput Sequencing

High-throughput sequencing was performed on total RNA extracted from UK*Ma*1 and PL*Mg*1 using Illumina MiSeq. Genomic DNA (gDNA) was depleted using DNaseI, and ribosomal RNA (rRNA) was removed using NEBNext^®^ rRNA Depletion Kit (Human/Mouse/Rat) with RNA Sample Purification Beads (New England Biolabs, Ipswich, Massachusetts, USA). The samples were run on an Illumina MiSeq platform, generating a total of 8,045,758 paired reads for PL*Mg*1 and 14,140,770 paired reads for UK*Ma*1. Each read length was 2 × 150 bp, and the insert size was 400 bp on average. All the Next Generation Sequencing (NGS) data were analysed using Geneious 11.1.2 software. The reads were assembled into contigs with de novo assembly using Geneious. The assembled contigs were compared against the RefSeq sequence database of all the virus proteins downloaded from GenBank using blastX. The minimum *e*-value was 1 × 10^−5^ to maintain high sensitivity and low rates of false hits [13]. The contigs that matched known viruses were verified by PCR.

### 2.4. Genome Retrieval with PCR

The primers used for primer walking were either designed based on the Lucheng Rn coronavirus (Genbank Acc. No. KF294380) reference sequence or based on the known genome sequences recovered from UK*Rn*3. The primers for contig confirmation and gap-filling between contigs for UK*Ma*1 and PL*Mg*1 were designed based on the sequences of the contigs. Geneious 11.1.2 software was used for genome mapping, annotating, and analysis. All the primers were evaluated in silico with the Primer3 online tool. For the retrieval of a short *ORF1b* fragment for L232, a previously described primer combination was used [9].

PCR reactions were carried out in a PCT-200 Peltier Thermal Cycler (MJ Research). The PCR reactions were performed with HotStarTaq Polymerase (Qiagen) according to manufacturer’s instructions.

The coronavirus sequences generated in this study were deposited in GenBank under the accession numbers MK163627 and MK249067-MK249069.

### 2.5. Phylogenetic Analysis

Reference sequences of α, β, and one γ-CoV were downloaded from GenBank for the *ORF1b*, *S*, and *N* genes and aligned using ClustalW [14] recognising codon positions. To provide evolutionary context, phylogenetic analyses were performed using the maximum likelihood (ML) method within the Molecular Evolutionary Genetics Analysis version 7 (MEGA7) [15] package. These analyses utilized a General Time Reversible (GTR) nucleotide substitution model with a gamma distribution of rate variation, a class of invariant sites (Γ+I) model of evolution, and complete deletion of gaps, with statistical robustness assessed using bootstrap resampling (1,000 pseudoreplicates). Equivalent ML phylogenies were also inferred using partial *ORF1b*, *S*, and *N* amino acid sequences and employing the Jones-Taylor-Thornton (JTT) substitution model with uniform rates, complete deletion of gaps, and statistical robustness of 1000 pseudoreplicates. Details regarding the number of sequences used for the phylogenetic analysis of each gene along with their length are given in Table 2, and the coronavirus reference sequences used in the phylogenetic analysis are given in Appendix A.

To determine the phylogenetic relationships among the host species analysed, the mitochondrial *cytochrome b* reference sequences of *Rattus*, *Apodemus*, *Myodes*, *Microtus* spp., and *Oryctolagus cuniculus* were downloaded from Genbank. In total, 44 full length (1136 nt) *cytochrome b* sequences were aligned as described above, and an ML tree was estimated using the same procedure as described above.

### 2.6. Analysis of the Virus-Host Co-Divergence

To compare the evolutionary history of rodents and their coronaviruses, particularly the relative frequencies of co-divergence versus cross-species transmission, a tanglegram was inferred by comparing the ML phylogenies of the rodent *cytochrome b* and the coronavirus *ORF1b* sequences, connecting hosts to their corresponding coronaviruses. The reconciliation between the alpha-CoV *ORF1b* and the host mtDNA *cytochrome b* phylogenies was performed using the Jane software with default parameters [16].

## 3. Results

### 3.1. Genome Retrieval of Rodent and Rabbit Alphacoronaviruses

The near-full genome sequence of the UK rat CoV (UK*Rn*3) was retrieved with primer-walking PCR. The 5′ and 3′ ends corresponded to the rat Lucheng CoV (KF294380)-derived primer sequences used for amplification of the genome termini and were not confirmed by the rapid amplification of cDNA ends (RACE) method. The genome of UK*Rn*3 had a length of 28,763 nt and 40.2% GC content. Its pairwise distance to Lucheng CoV at the nucleotide level was 95.5%, indicating that they belong in the same species. Additionally, the rat UK*Rn*3 was 90.2% similar to RtMruf (KY370045), 95.4% to RtRl-CoV (KY370050), and 75% to AcCoV-JC34 (NC034972) in the nucleotide level.

For the genome retrieval of the UK field vole CoV (UK*Ma*1) and the Poland-resident bank vole (PL*Mg*1), primer-walking PCR was not applicable due to the lack of a very close reference sequence. Thus, NGS performed on total RNA extracted from the *Microtus agrestis* gut and *Myodes glareolus* liver was used as an alternative method. Genome fragments representing 17,483 and 8603 nucleotides spanning various genes were retrieved for UK*Ma*1 and PL*Mg*1, respectively. Full-length *N* gene and partial *ORF1ab*, *ORF2*, *S*, *ORF6*, *ORF8*, and *ORF9* genes were retrieved for *UKMa*1, whereas partial *ORF1b* and *N* genes were retrieved for PL*Mg*1. For the L232 coronavirus, which was obtained from a rabbit living in France [12], additional sequences up to 621 bp of the *ORF1b* gene were retrieved by targeting conserved regions with PCR. Unfortunately, presumably due to the relatively poor quality of the recovered nucleic acid, we were unable to recover additional sequences.

### 3.2. Virus Genome Characterization

Genome analysis of all the rodent alpha-CoVs revealed a number of common features (Figure 1). All viruses, with the exception of the field mouse AcCoV-JC34 (NC034972), possessed a predicted *ORF2* gene that was approximately 274 amino acids (aa) in length and was located between the *ORF1b* and *S* gene. Blast search revealed that this gene was not present in any other characterized alpha-CoVs and its closest match, at 44% amino acid similarity, was the NS2a gene of the Betacoronavirus1 Human CoV OC43. Interestingly, all the rodent α-CoVs possessed a β-CoV-like spike, which indicates recombination between α and β-CoV. All the rodent α-CoVs possessed a predicted 214 aa long *ORF3* gene located between the *S* and the *E* gene. The closest match for *ORF3* was the 3c-like protein of ferret alphacoronavirus with 47% similarity at the amino acid level. Another striking feature of rodent alphacoronavirus genomes was the presence of the predicted *ORF6* (165 aa) gene between the *M* and *N* genes, which is observed in no other members of the genus *Alphacoronavirus*. Other common features of all the rodent α-CoV were the presence of a predicted *ORF8* (~181 aa) within the *N* gene and *ORF9* (~111 aa) downstream of the *N* gene. Finally, the grey red-backed vole RtMruf-CoV (KY370045) appeared to possess an additional 154 aa long predicted gene called *ORF2b*, located between *ORF2* and *S*.

Further analysis revealed similarities in the core part of the transcription regulatory sequence (TRS) located upstream of the 5′ end of each ORF (Table 3). All the rodent α-CoVs possessed the same 5′-AACUAA-3′ core TRS sequence in the majority of their genes. However, some variations of the core TRS were observed; specifically, a 5′-AACUUUAA-3′ for *ORF2*, 5′-UACUAA-3′ for *ORF3*, 5′-CACUAA-3′, and 5′-GACUAA-3′ for *ORF8*.

### 3.3. Phylogenetic Analysis of Rodent Alphacoronaviruses Suggests a Shared Common Ancestry

Having retrieved the whole genome for UK*Rn*3 and partial genomes for UK*Ma*1, PL*Mg*1, and L232, we next evaluated the evolutionary relationships between the novel α-CoVs and the coronavirus sequences publicly available on GenBank. Accordingly, an analysis of a conserved 576 bp region of the *ORF1b* gene, which encodes the RNA-depended-RNA-polymerase (RdRp), revealed that all the rodent alpha-CoVs sampled from geographic localities as diverse as Western Europe and East Asia, as well as the rabbit L232 CoV, formed a distinct clade within the *Alphacoronavirus* genus with 100% bootstrap support (Figure 2). Within that cluster, viruses obtained from UK and Chinese rat species formed a single sub-clade with 100% bootstrap support. Other well supported subclades (i.e., all supported by >80% bootstrap replicates) contained (i) the Chinese grey red-backed vole (RtMruf), the German bank vole (RCoV/RMU10 3212/Myo gla/GER/2010), the Chinese field mouse (AcCoV-JC34), and (ii) the UK field vole (UK*Ma*1) and French rabbit CoV (L232). The virus sampled from a Poland-resident bank vole (PL*Mg*1) formed a distinct lineage.

Phylogenetic analysis of the *N* gene (Figure 3) revealed similar results to the *ORF1b* phylogeny, with all the available rodent coronaviruses forming a distinct clade within the genus *Alphacoronavirus*. The rat viruses contained within a single cluster supported by 100% of bootstrap trees, while PL*Mg*1, UK*Ma*1, and AcCoV-JC34 fell in strongly supported more divergent positions. The phylogenetic relationships within the rodent clade are therefore the same as those observed in the *ORF1b* phylogeny (rabbit L232 is not shown because it was not possible to retrieve the corresponding *N* gene sequence).

A markedly different evolutionary relationship was observed in the case of the partial *S* gene phylogeny, in which all the rodent-borne CoVs previously assigned as alpha-CoVs based on their *ORF1* and *N* sequences clustered within the genus *Betacoronavirus* (Figure 4). Within this genome region, these viruses clustered with other known recombinant coronaviruses, including Wencheng Sm shrew CoV, Bat CoV HKU2, BtRf-AlphaCoV, Swine acute diarrhoea syndrome CoV, and swine acute diarrhoea syndrome related (bat) CoV. The rodent coronaviruses again formed a single branch with 100% bootstrap support, indicative of an ancient recombination event between the α- and β CoVs.

Importantly, amino acid-based ML trees were also inferred for the *ORF1b*, *N*, and *S*, and exhibited the same clustering of rodent alpha-CoVs with high bootstrap support (not shown, available on request).

### 3.4. Comparison of Virus and Host Phylogenies

To determine the extent to which the rodent CoVs may have co-diverged with their rodent hosts, indicative of a very long-term virus-host association, we examined the extent of congruence between the virus *ORF1b* (with the same topology observed in the *N* gene) and host mitochondrial (mt) *cytochrome b* phylogenies by inferring tanglegrams (Figure 5A). The tanglegram was generated by linking the phylogenies of rodent alpha-CoV *ORF1b* to their equivalent rodent species and assessing whether there were changes in the branching order in the two phylogenies, which would indicate cross-species transmission. The rabbit (*Oryctolagus cuniculus*; order Lagomorpha) sequence was used as outgroup as it has previously been shown to be closely related to the *Rodentia* order [5]. Notably, the topology of the *cytochrome b* tree did not fully match that observed for the rodent alpha-CoVs, in which PL*Mg*1 (*Myodes* spp.) fell basal to all the other viruses, and RtMruf and RCoV/RMU10 3212/Myo gla/GER/2010 (*Myodes* spp.) were more closely related to *Rattus* spp. coronaviruses (UK*Rn*3, Lucheng Rn, RtRl). Moreover, the rodent UK*Ma*1 (*Microtus* spp.) clustered with L232 (*Oryctolagus cuniculus*) in the *ORF1b* tree, again indicative of some cross-species virus transmission.

In addition to the tanglegram, we conducted a co-phylogenetic analysis of the different types of evolutionary events that likely shaped the evolution of the rodent CoVs with their mammalian hosts (mtDNA *cytochrome b*) (Figure 5B). The co-phylogenetic JANE analysis between the rodent alpha-CoV *ORF1b* and the rodent *cytochrome b* phylogenies was performed to assess whether the two phylogenies matched, which would be indicative of co-evolution of virus and host. Under the most likely co-phylogenetic scenario, virus-host co-divergence explained 22% of all evolutionary events, with host-switching (i.e., cross-species transmission) more frequent at 45% (and duplications accounting for 33%), although the relative frequency of these events is likely to change with more extensive sampling. Hence, these data suggest that although the rodent alpha-CoV formed a monophyletic group indicative of a long virus-host association, the evolution of this group of viruses was likely characterized by relatively frequent cross-species transmission.

## 4. Discussion

We [9] and others [10,11,12,17,18] have previously shown the presence of novel CoVs in a range of rodent species. Here, we show that all rodent alpha-CoVs sampled to date—and from West Europe and East Asia—form a monophyletic group and have similar genome structure and common gene constellations and a recombinant spike gene. Hence, these data suggest that the rodent alpha-CoVs sampled to date originate from a single common ancestor and that there is a long association between rodents and CoVs. Moreover, our co-phylogenetic analysis revealed that the phylogenetic diversity in the rodent group was likely shaped by a combination of virus-host divergence and cross-species transmission, which appears to be a common mode of macroevolution in RNA viruses [19].

Also of note was the presence of a recombinant *Betacoronavirus* spike gene that clustered with a number of previously reported recombinant viruses that infect a diverse array of mammalian species, one of which was associated with a recent fatal outbreak in pigs in China [20,21,22,23]. Hence, the recombination event involving the *S* gene clearly occurred early on in the evolutionary history of the alpha- and betacoronaviruses. 

In addition to the presence of a *Betacoronavirus* spike, the rodent CoVs also had similar, but not identical, gene constellations and genome features. Again, the most parsimonious explanation of this would be that all the rodent alpha-CoVs sampled to date originate from a single common ancestor, and the lack of an *ORF2* (complete deletion) in AcCoV-JC34 likely reflects a secondary evolutionary loss.

Further analysis showed that the rodent alpha-CoVs shared the same predicted core TRS sequences upstream to each gene with a few exceptions. The predominant core TRS was 5′-AACUAA-3′. The same core sequence was also observed in Human NL63 (5′-AACUAAA-3′) [24] and Bat HKU2 CoV (5′-AACUAAA-3′) [20]. Although it was mentioned that no TRS was detected for *ORF3* in Lucheng Rn CoV [17], we used the core TRS of AcCoV-JC34 (5′-UACUAAA-3′) [10] for *ORF3* as a reference, and we were able to find the same sequence in Lucheng Rn CoV, UK*Rn*3, and RtMruf. Other variations such as 5′-CACUAA-3′ and 5′-GACUAA-3′ were observed in *ORF8* of UK*Ma*1 and RtMruf, respectively. The variation 5′-CUAAAC-3′ has been observed in Feline infectious peritonitis virus (FIPV) and Transmissible gastroenteritis virus (TGEV) [25,26]. The conserved core TRS among rodent alpha-CoVs is another indicator that they originate from a common ancestor.

Comparison of rodent species’ *cytochrome b* and coronavirus *ORF1b* evolution showed incomplete congruence, such that despite forming a monophyletic group, these rodent CoVs also experienced cross-species transmission. In particular, according to the *cytochrome b* analysis, *Rattus* spp. formed a distinct clade with *Apodemus* spp., whereas *Microtus* spp. formed a separate clade with *Myodes* spp. However, according to *ORF1b* phylogeny, *Rattus* spp. CoVs clustered with two *Myodes* spp. CoV (RtMruf and RCoV/RMU10 3212/Myo gla/GER/2010), suggesting cross-species transmission (and this was confirmed in the co-phylogenetic analysis). Cross-species transmission is frequently observed in coronaviruses and representative examples are SARS-CoV, MERS-CoV, and SADS/SADSr-CoV [21,27].

That the rodent alpha-CoVs form a monophyletic group despite being sampled from multiple localities suggests that they have been associated with rodents for an extended period, although providing an exact chronological context is complex. *Rattus norvegicus* originated in Southeast Asia and spread to Northeast Asia approximately 200,000 years ago. It then spread from Southeast Asia to the Middle East ~3100 years ago, then to Africa ~2000 years ago, and finally to Europe ~1800 years ago [28]. Therefore, it is possible that even though the European UK*Rn*3 coronavirus appears to be approximately as old as Lucheng *Rn* CoV, the global spread of the brown rat implies that the virus introduction into Europe occurred with that host and was a relatively recent event.

In sum, we show that all the rodent alpha-CoVs described to date share a single common ancestor and have likely evolved through a combination of virus-host co-divergence and cross-species transmission. Not only is this common ancestry evident in the phylogenetic relationships inferred from multiple genes, but also genomic signatures are the recombinant spike gene, same gene constellations (*ORF2*, *ORF3*, *ORF6*, *ORF8*, and *ORF9*), and similar core TRS sequences. Finally, although the current sequence data set of rodent alphacoronaviruses is relatively small, it is representative of a number of different rodent species living across a wide geographical area.

## Figures and Tables

**Figure 1 viruses-11-00125-f001:**
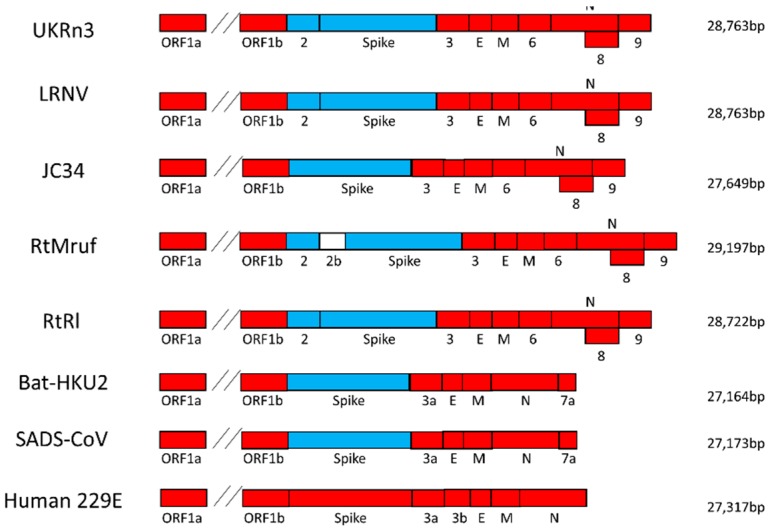
Genome organisation of rodent alphacoronaviruses. Schematic representation of the UK*Rn*3 genome organisation along with those of other rodent alphacoronaviruses (α-CoVs): Lucheng Rn CoV (LRNV), AcCoV-JC34, RtRl-CoV, RtMruf-CoV, and the related Bat-HKU2 and Swine acute diarrhoea syndrome coronaviruses (CoV) (SADS-CoV). Its genome organisation was similar to those of members of the genus *Alphacoronavirus*, with the characteristic 5′- *ORF1ab* - *spike (S)* - *envelope (E)* – *membrane (M)* - *nucleocapsid (N)* - 3′ gene order. The red colour represents genes that belong to the genus *Alphacoronavirus*, whereas light blue denotes genes that belong to the genus *Betacoronavirus*, indicative of recombination. The Human 229E genome represents a typical alpha-CoV genome organisation.

**Figure 2 viruses-11-00125-f002:**
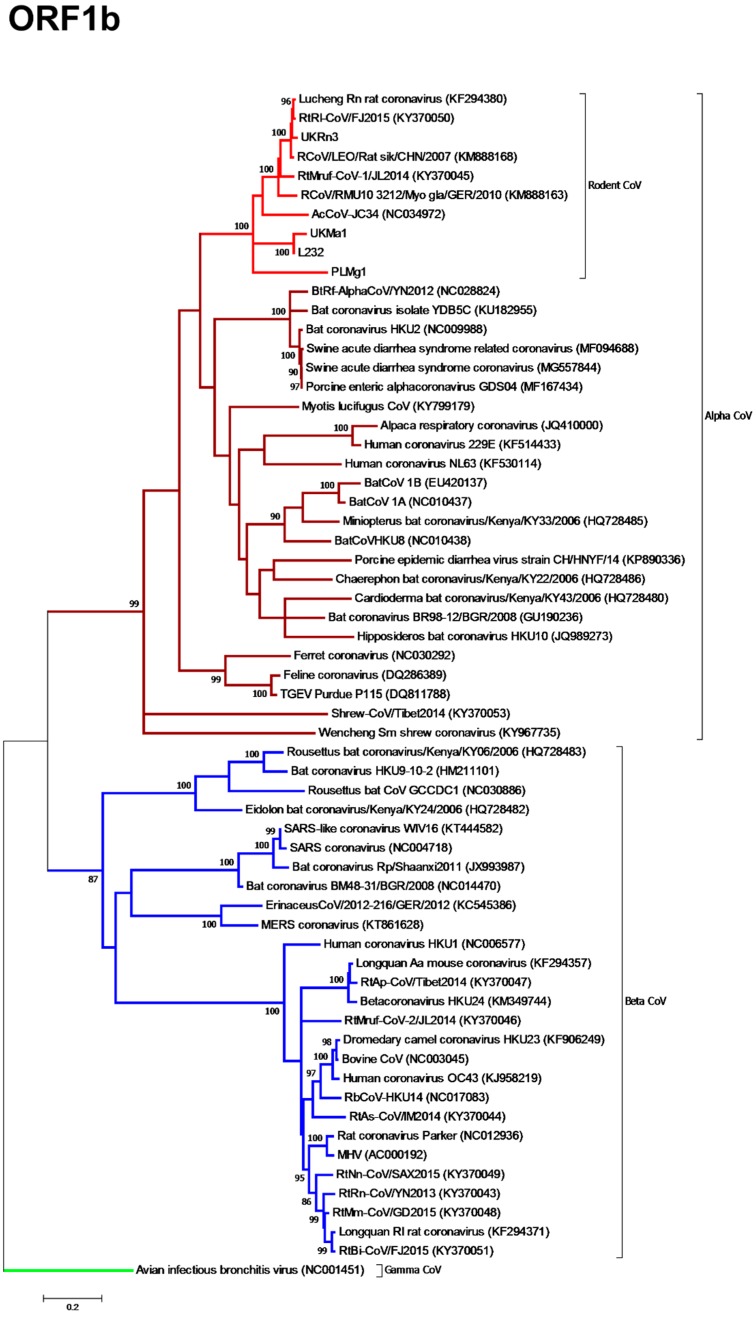
Phylogenetic relationships of the rodent alphacoronaviruses based on the *ORF1b* gene. Maximum likelihood phylogenetic analysis of coronavirus partial 576 nt *ORF1b* gene sequences, revealing that the rodent alphaviruses formed a single clade with the novel rabbit L232 CoV within the genus alphacoronavirus. The novel coronavirus sequence obtained from a French rabbit L232 CoV was analysed alongside reference sequences representing three different coronavirus genera (*Alphacoronavirus*, *Betacoronavirus*, and *Gammacoronavirus* as an outgroup). Reference sequences are indicated by their GenBank accession numbers. Branch lengths are drawn to a scale of nucleotide substitutions per site. Numbers above individual branches indicate the percentage that that branch was found in 1000 bootstrap replicates; only values >80% are shown. Alpha-CoVs are highlighted in brown, betacoronaviruses in blue, gammacoronaviruses in green, and the rodent coronaviruses in light red.

**Figure 3 viruses-11-00125-f003:**
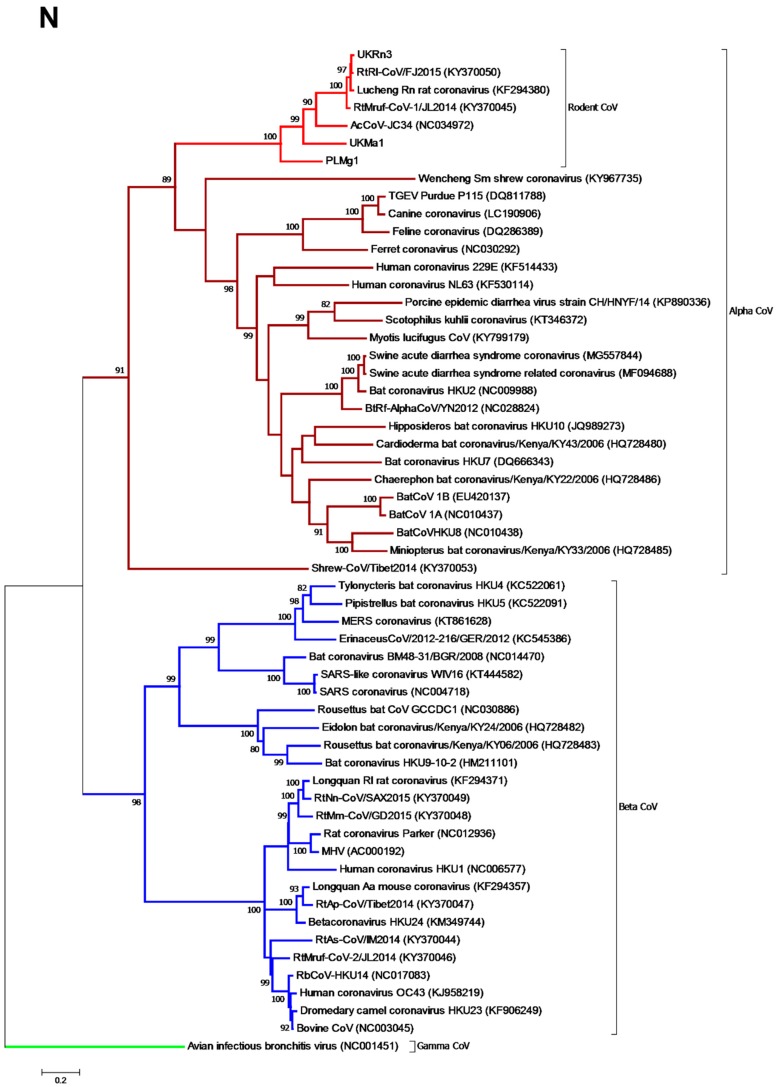
Phylogenetic relationships of the rodent alphacoronaviruses based on the *N* gene. Maximum likelihood phylogenetic analysis of coronavirus partial *N* gene sequences corresponding to positions 10–1173 of the Lucheng Rn CoV (KF294380). Novel coronavirus sequences obtained from PL-resident Myodes glareolus (PL*Mg*1) and the UK *Microtus agrestis* (UK*Ma*1) and the UK *Rattus norvegicus* (UK*Rn*3) were analysed alongside reference sequences representing three different coronavirus genera (*Alphacoronavirus*, *Betacoronavirus*, and *Gammacoronavirus* as an outgroup). Reference sequences are indicated by their GenBank accession numbers. Branch lengths are drawn to a scale of nucleotide substitutions per site. Numbers above individual branches indicate the percentage that that branch was found in 1000 bootstrap replicates; only values >80% are shown. Alpha-CoVs are highlighted in brown, betacoronaviruses in blue, gammacoronaviruses in green, and the rodent coronaviruses in light red.

**Figure 4 viruses-11-00125-f004:**
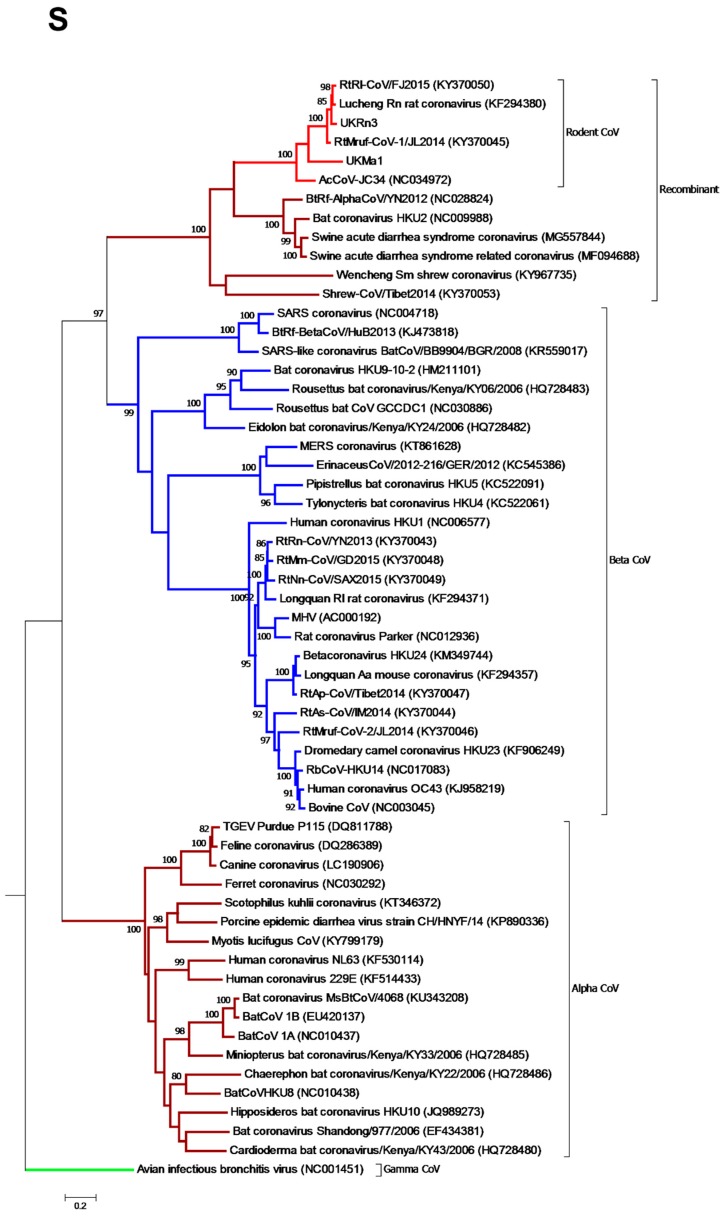
Phylogenetic relationships of the rodent alphacoronaviruses based on the *S* gene. Maximum likelihood phylogenetic analysis of coronavirus partial *S* gene sequences corresponding to positions 1831–3228 of the Lucheng Rn CoV (KF294380), indicating that the rodent alpha-CoVs formed a distinct clade within the genus *Betacoronavirus*. The novel coronavirus sequence obtained from a UK-resident *Microtus agrestis* was analysed alongside reference sequences representing three different coronavirus genera (*Alphacoronavirus*, *Betacoronavirus*, and *Gammacoronavirus* as an outgroup). Reference sequences are indicated by their GenBank accession numbers. Branch lengths are drawn to a scale of nucleotide substitutions per site. Numbers above individual branches indicate the percentage that the branch was found in 1000 bootstrap replicates; only values >80% are shown. Alpha-CoVs are highlighted in brown, betacoronaviruses in blue, gammacoronaviruses in green, and the rodent coronaviruses in light red.

**Figure 5 viruses-11-00125-f005:**
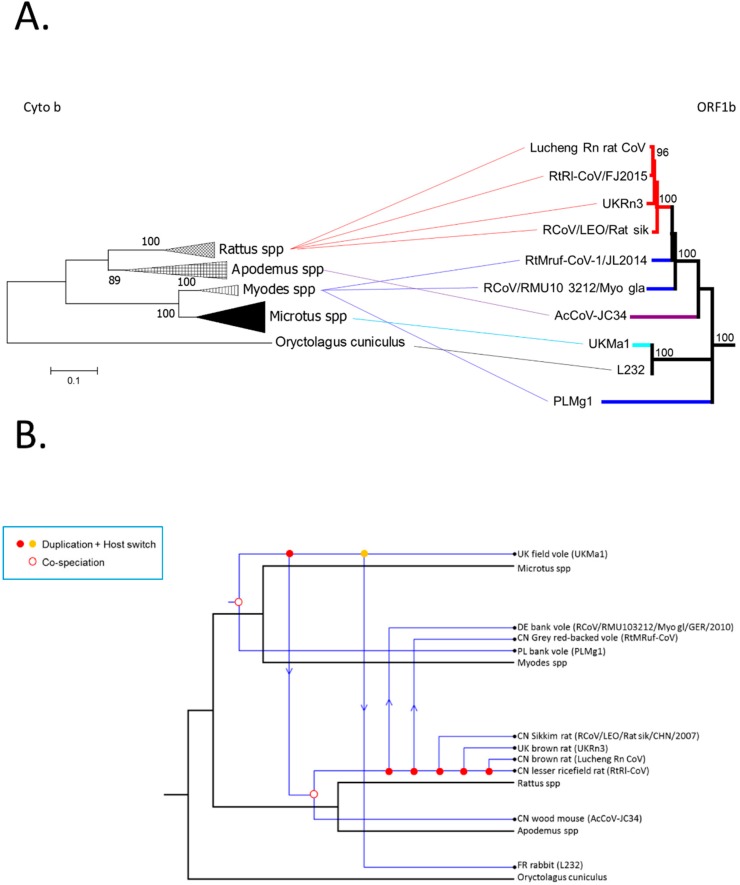
(**A**) Tanglegram of rodent *cytochrome b* and CoV *ORF1b* evolution. Maximum likelihood phylogenetic analysis of full-length rodent *cytochrome b* nucleotide sequences compared to rodent alpha-CoV *ORF1b* phylogeny. *Cytochrome b* sequences were obtained from GenBank for the rodent species *Microtus*, *Myodes*, *Rattus*, and *Apodemus*, and the rabbit *Oryctolagus cuniculus*, which was used as an outgroup. Branch lengths are drawn to scale: the bar indicates 0.1 nucleotide substitutions per site. Numbers above individual branches indicate the percentage that that branch was found in 1000 bootstrap replicates; only percentages >80 are shown. Blue indicates the *Myodes* spp., cyan the *Microtus* spp., purple the *Apodemus* spp., red *Rattus* spp., and black indicates *Oryctolagus cuniculus*. (**B**) Reconciliation of the sampled rodent alpha-CoV phylogeny with that of their mammalian hosts, utilizing the co-phylogenetic method implemented in the Jane package [16]. This figure shows the maximum possible co-divergence, duplication, and host switching events.

**Table 1 viruses-11-00125-t001:** List of sample name, species, country of origin, organ, and original publication.

Name	Species	Country	Organ	Reference
**UK*Rn*3**	*Rattus norvegicus*	United Kingdom	Liver	[9]
**UK*Ma*1**	*Microtus agrestis*	United Kingdom	Gut	[9]
**PL*Mg*1**	*Myodes glareolus*	Poland	Liver	[9]
**L232**	*Oryctolagus cuniculus*	France	Intestinal fluid	[12]

**Table 2 viruses-11-00125-t002:** List of the different genes used for phylogenetic analyses, number of sequences, sequence length, and corresponding positions on the Lucheng CoV genome.

Gene	Number of Sequences	Length (nt)	Corresponding Positions on Lucheng CoV Genome
***ORF1b***	46	576	14,140–14,715
***S***	31	1398	23,242–24,639
***N***	32	1170	26,977–28,146

**Table 3 viruses-11-00125-t003:** List of all the predicted core transcription regulatory sequences (TRS) for all the genes of the rodent alpha-CoVs. Core TRS could not be predicted for ORF1ab, S, ORF3, E, and M for UKMa1 as there was no available sequence for those genes. AcCoV-JC34 did not possess an ORF2 such that there was no core TRS for that gene.

Virus	Genes									
	*ORF1ab*	*ORF2*	*S*	*ORF3*	*E*	*M*	*ORF6*	*N*	*ORF8*	*ORF9*
**UK*Rn*3**	AACUAA	AACUUUAA	AACUAA	UACUAA	AACUAA	AACUAA	AACUAA	AACUAA	AACUAA	AACUAA
**LRNV**	AACUAA	AACUAA	AACUAA	UACUAA	AACUAA	AACUAA	AACUAA	AACUAA	AACUAA	AACUAA
**UK*Ma*1**	N/A	AACUUUAA	N/A	N/A	N/A	N/A	AACUAA	AACUAA	CACUAA	AACUAA
**RtRl**	AACUAA	AACUUUAA	AACUAA	AACUAA	AACUAA	AACUAA	AACUAA	AACUAA	AACUAA	AACUAA
**RtMruf**	AACUAA	AACUUUAA	AACUAA	UACUAA	AACUAA	AACUAA	AACUAA	AACUAA	GACUAA	AACUAA
**JC34**	AACUAA	N/A	AACUUA	UACUAAA	AACUAA	AACUAA	AACUAA	AACUAA	CACUAA	AACUAA

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
