# Peer review of "Shared Common Ancestry of Rodent Alphacoronaviruses Sampled Globally"

_viruses, 2019, doi:10.3390/v11020125_

Reviewer 1 Report

Dear Authors,

Thank you for submitting this interesting paper detailing the shared common ancestry of rodent alphacoronavirus. It was well written for the most part easy to understand. Nice job.

The title is a bit misleading. The paper is really about the shared common ancestry or rodent alphacoronavirus. The title sounds as though you are detailing the shared common ancestry of all alphacoronavirus that originated in rodents. So, I would suggest rearranging the words in the title a bit to more accurately reflect the data in the paper.

Line 80. Please detail or cite the method for depleting rRNA.

Figure 1. This figure could be improved by also including a prototypic alphacoronavirus (PDEV, 229E, etc.) as a comparator. Also, the explanation as to why spike is believed to have originated from betacoronavirus should be mentioned here in some way or perhaps reference this will be discussed in a later figure. It seems odd that many other proteins as discussed in the text tied to Figure 1, yet spike is not. The blue coloring of spike and the statement in the legend "whereas light blue denotes genes that belong to the genus Betacoronavirus, indicative of recombination" led me to believe that this statement would be substantiated by additional text in the body of the paper. 

Line 180. I am interested to know what proteins are encoded in the 576pb region of ORF1b used to build the NJ Tree.

Line 206. "L232 sequence is absent." Without explanation, this sounds odd. Some digging in the methods leads me to believe that only ORF1b sequence was recovered for the rabbit virus. This the sequence was not included in the N gene NJ tree. Please clarify.

Line 243. A brief description of tanglegrams, how they work and what they tell us would greatly help a general virology audience understand how and why this analysis was performed.

Line 254. SImilar to the tanglegram, a brief explanation of the co-phylogenetic analysis would greatly improve this section for clarity for the general virology audience.

Overall, a cool paper and a valuable addition to our current understanding of zoonotic coronavirus ecology. 

Author Response

 Thank you for considering our manuscript for publication in Viruses and also we would like to extend our thanks to the reviewers for their thoughtful suggestions and comments.

 I can confirm that the revised manuscripts has incorporated the suggested changes and also clarified the raised queries as follows:

 For reviewer 1:

·       “The title is a bit misleading. The paper is really about the shared common ancestry or rodent alphacoronavirus. The title sounds as though you are detailing the shared common ancestry of all alphacoronavirus that originated in rodents. So, I would suggest rearranging the words in the title a bit to more accurately reflect the data in the paper.”

– Title changed to “Shared common ancestry of rodent alphacoronaviruses sampled globally”

·       Line 80. Please detail or cite the method for depleting rRNA.

-       We added: “..and ribosomal RNA (rRNA) was removed using NEBNext® rRNA Depletion Kit (Human/Mouse/Rat) with RNA Sample Purification Beads (New England Biolabs).”

·       Figure 1. This figure could be improved by also including a prototypic alphacoronavirus (PDEV, 229E, etc.) as a comparator. Also, the explanation as to why spike is believed to have originated from betacoronavirus should be mentioned here in some way or perhaps reference this will be discussed in a later figure. It seems odd that many other proteins as discussed in the text tied to Figure 1, yet spike is not. The blue coloring of spike and the statement in the legend "whereas light blue denotes genes that belong to the genus Betacoronavirus, indicative of recombination" led me to believe that this statement would be substantiated by additional text in the body of the paper.

-       We modified Figure 1 by adding Human 229E as a reference alpha-CoV, as suggested by the reviewer. Also, in line 155 we added: “Interestingly, all the rodent alpha-CoVs possessed a beta-CoV-like spike which indicates recombination between alpha and beta-CoV.

-       In line 178 in the caption of Figure1 we added: “The Human 229E genome represents a typical alpha-CoV genome organization”.

·       Line 180. I am interested to know what proteins are encoded in the 576pb region of ORF1b used to build the NJ Tree.

-       In line 189 we added: “…which encodes the RNA-depended-RNA-polymerase (RdRp)…”

·       Line 206. "L232 sequence is absent." Without explanation, this sounds odd. Some digging in the methods leads me to believe that only ORF1b sequence was recovered for the rabbit virus. This the sequence was not included in the N gene NJ tree. Please clarify.

-       In line 215 we added: “…(Rabbit L232 is not shown because it was not possible to retrieve the corresponding N gene sequence)..”

·       Line 243. A brief description of tanglegrams, how they work and what they tell us would greatly help a general virology audience understand how and why this analysis was performed.

-       In line 258 we added: “The tanglegram was generated by linking the phylogenies of rodent alpha-CoV ORF1b to their equivalent rodent species and assessing whether there were changes in the branching order in the two phylogenies, which would indicate cross-species transmission.”

·       Line 254. Similar to the tanglegram, a brief explanation of the co-phylogenetic analysis would greatly improve this section for clarity for the general virology audience.

In line 270 we added: “The co-phylogenetic JANE analysis between the rodent alpha-CoV ORF1b and the rodent cytochrome b phylogenies was performed to assess whether the two phylogenies matched, which would be indicative of co-evolution of virus and host.” 

Reviewer 2 Report

"Shared common ancestry of alphacoronaviruses in globally distributed rodents" by Tsoleridis et al was well-written, the conclusion was supported by their analysis. Although the discovery of a rodent CoV was not novel, the authors showed their data in a different way and thus found rodent alpha-CoVs shared common ancestry. The paper should be interesting to CoV virologists. A couple of minor issues:

1, there is no new rabbit CoV been sequenced. Please confirm in the first paragraph of results part.

2, what is the genome identity between the new rat CoV and other rodent CoVs?

3, the author claimed rodent S gene was from beta-CoV, but they didn't do analysis in the paper, at least for the new CoV. An evidence should be shown.

Author Response

Thank you for considering our manuscript for publication in Viruses and also we would like to extend our thanks to the reviewers for their thoughtful suggestions and comments.

 I can confirm that the revised manuscripts has incorporated the suggested changes and also clarified the raised queries as follows:

1.     There is no new rabbit CoV been sequenced. Please confirm in the first paragraph of results part.

-In line 146 we added: “…additional sequence up to…”. As in our study we extended the sequence read available for the ORF1b.

 2.     What is the genome identity between the new rat CoV and other rodent CoVs?

-In line 137 we added: “Additionally, the rat UKRn3 was 90.2% similar to RtMruf (KY370045), 95.4% to RtRl-CoV (KY370050) and 75% to AcCoV-JC34 (NC034972) at the nucleotide level.”

 3.     The author claimed rodent S gene was from beta-CoV, but they didn't do analysis in the paper, at least for the new CoV. An evidence should be shown.

-We justify our claim with the phylogenetic analysis of the S gene where the rodent alpha-CoVs cluster within the beta-CoV genus. The description is detailed in lines 230-237: “A markedly different evolutionary relationship was observed in the case of the partial S gene phylogeny, in which all the rodent-borne CoVs previously assigned as alpha-CoVs based on their ORF1 and N sequences clustered within the genus Betacoronavirus (Figure 4). Within this genome region, these viruses clustered with other known recombinant coronaviruses including Wencheng Sm shrew CoV, Bat CoV HKU2, BtRf-AlphaCoV, Swine acute diarrhoea syndrome CoV and swine acute diarrhoea syndrome related (bat) CoV. The rodent coronaviruses again formed a single branch with 100% bootstrap support, indicative of an ancient recombination event between the alpha- and beta CoVs.”

Yours faithfully,

Jonathan K. Ball

Professor of Molecular Virology
